

# Ability to predict repetitions to momentary failure is not perfectly accurate, though improves with resistance training experience

James Steele[1], Andreas Endres[2], James Fisher[1], Paulo Gentil[3] and Jürgen Giessing[2]

[1] School of Sport, Health, and Social Sciences, Southampton Solent University, Southampton, United Kingdom
[2] Institute of Sport Science, University of Koblenz-Landau, Landau, Germany
[3] Faculty of Physical Education and Dance, Federal University of Goias, Goiânia, Brazil

## ABSTRACT

'Repetitions in Reserve' (RIR) scales in resistance training (RT) are used to control effort but assume people accurately predict performance *a priori* (i.e. the number of possible repetitions to momentary failure (MF)). This study examined the ability of trainees with different experience levels to predict number of repetitions to MF. One hundred and forty-one participants underwent a full body RT session involving single sets to MF and were asked to predict the number of repetitions they could complete before reaching MF on each exercise. Participants underpredicted the number of repetitions they could perform to MF (Standard error of measurements [95% confidence intervals] for combined sample ranged between 2.64 [2.36–2.99] and 3.38 [3.02–3.83]). There was a tendency towards improved accuracy with greater experience. Ability to predict repetitions to MF is not perfectly accurate among most trainees though may improve with experience. Thus, RIR should be used cautiously in prescription of RT. Trainers and trainees should be aware of this as it may have implications for the attainment of training goals, particularly muscular hypertrophy.

## INTRODUCTION

Resistance training (RT) is an exercise mode evidenced to provide a wide range of health benefits (*Steele et al., 2017c*). As such, understanding prescription of RT to maximise these outcomes is of considerable interest. One variable which may be of primary importance is the intensity of effort employed i.e., whether or not RT is performed to momentary failure (MF; *Steele, 2014*; *Steele et al., 2017a*). A plethora of recent work shows that when effort is matched by having RT performed to MF, manipulations of other RT variables have a lesser impact upon the magnitude of outcomes. These secondary variables include; load (*Morton et al., 2016*; *Schoenfeld et al., 2015*; *Schoenfeld et al., 2016*; *Fisher, Ironside & Steele, 2016*), repetition duration (*Schoenfeld, Ogborn & Krieger, 2015*), and the use of advanced or complicated training methods such as pre-exhaustion (*Fisher et al., 2014*),

Corresponding author
James Steele,
james.steele@solent.ac.uk

breakdown sets (*Fisher, Carlson & Steele, 2015*) or occlusion training (*Barcelos et al., 2015*; *Farup et al., 2015*).

A number of recent reviews have also concluded that training to MF may confer greater adaptations in strength (*Fisher et al., 2011*), hypertrophy (*Fisher, Steele & Smith, 2013*) and possibly cardiorespiratory fitness (*Steele et al., 2012*) than training not to MF. Conversely, more recent empirical work has shown contrasting results regarding the efficacy of training to MF (*Fisher, Blossom & Steele, 2016*; *Giessing et al., 2016a*; *Giessing et al., 2016b*; *Izquierdo-Gabarren et al., 2010*; *Sampson & Groeller, 2016*). Proximity to MF has been considered a determinant of the effort employed during RT, and MF (as a set end-point) has been suggested as the only way to objectively match inter- and intra-individual effort due to the variations in number of repetitions possible prior to MF at the same relative loads i.e., % one repetition maximum (%1RM; *Steele, 2014*; *Steele et al., 2017a*). In fact, it has been argued that training to MF is the most appropriate way to control the application of a RT stimulus (*Dankel et al., 2016*).

However, the current body of research has typically considered this variable dichotomously (i.e., people training 'to MF' or 'not to MF'). As a result, the dose–response nature of sub maximal intensities of effort resulting from set end-points occurring at different proximities to MF is unclear. Furthermore, it is unclear whether there is a threshold of relative effort which optimises adaptations. As such, to understand sub maximal effort some have developed scales to assess effort during RT relative to MF (*Hackett et al., 2012*; *Hackett et al., 2016*; *Zourdos et al., 2016*; *Helms et al., 2016*). These 'Repetitions in Reserve' (RIR) scales are designed as a way of assessing/controlling relative effort by participants estimating how many repetitions they can perform before reaching MF.

In comparison with traditional rating of perceived effort (RPE) scales, RIR scales appear more likely to offer valid representations of effort when training to, or close to, MF (*Helms et al., 2016*) whereas traditional RPE often yields far less accurate ratings under such conditions (*Hackett et al., 2012*). Indeed, even when training to MF, traditional RPE is often less than maximal (*Steele et al., 2017b*). This in combination with the considerable inter- and intra-individual variations in number of repetitions possible prior to MF at the same relative loads suggests that RIR scales may offer an improvement in control of effort during RT compared with either use of %1RM or traditional RPE.

However, the use of these scales assumes that trainees are able to accurately predict the number of repetitions they could perform to MF with a particular load. Studies which have shown differences between groups training to MF compared with those not training to MF (where the participants were instructed to stop at the point they predicted MF on the next repetition), might be explained by the participants' inability to accurately predict MF (i.e., they actually stopped >1 repetitions away from MF; *Giessing et al., 2016a*; *Giessing et al., 2016b*). A study by *Hackett et al. (2012)* supports this, and revealed that trained participants were not perfectly accurate at predicting the number of repetitions they could perform to MF using a RIR scale, although their accuracy improved with subsequent sets. This would question the value of using 'intuitive' approaches to RT such as the RIR scales. Ability to predict proximity to MF may improve with training experience and thus it has been noted that the use of RIR scales may present greater value in experienced trainees

(*Helms et al., 2016*). However, a follow-up study from *Hackett et al. (2016)* study suggested that previous RT experience did not affect ability to predict proximity to MF.

In light of the potential value of training to MF, as well as the introduction of recent RIR scales to control RT effort, it is of interest to examine trainees with differing levels of RT experience in their ability to predict the number of repetitions they can perform to MF. Further, examination of this in controlled yet ecologically valid conditions such as their usual gym environment offers considerable practical information. As such the aim of this study was to compare predicted with actual repetitions to MF in participants with a range of RT experience.

## METHODS

### Experimental approach to the problem

Participants in this study underwent a single RT trial in order to examine whether they were able to accurately predict the number of repetitions they could perform when training to MF. All participants, grouped according to their RT experience, were asked to provide a prediction of repetitions to MF and then undergo a test of actual repetitions to MF for comparison.

### Participants

One hundred and forty-one participants (males $n = 72$, age $29 \pm 10$ years; females $n = 69$, age $25 \pm 8$ years) were recruited from the existing membership pool of a private exercise facility in Germany. Participants were required to have no medical condition for which RT would be contraindicated, and were grouped based upon duration of previous RT experience; <1.5 months (orientation, $n = 15$), 1.5 to six months (beginner, $n = 21$), six to 12 months (experienced, $n = 21$), 12 to 36 months (advanced, $n = 42$), and >36 months (expert, $n = 42$). Written informed consent was provided by all participants and the study was ethically approved by the author's institution.

### Procedures

Participants underwent a single RT session involving the following exercises: seated row, chest press, leg press, elbow flexion, and pulldown, all using selectorised resistance machines, and sit-ups using additional free weight loading. All participants were required to have been performing these exercises in their pre-existing training programs and to have the current training load they were using recorded in their training logs. Participants performed a single set of each exercise to concentric MF according to recent definitions of this concept i.e., the set ending when the trainee reached the point where, despite attempting to do so, they could not complete the concentric portion of their current repetition without deviation from the prescribed form of the exercise (*Steele et al., 2017a*). Participants were informed to use the repetition duration they normally used during training for each exercise, to retain familiarity. Exercises were performed in the order that the participants typically performed them in their current training based upon their recorded training logs and participants were permitted to rest between each exercise for as long they typically would or felt necessary to ensure maximal performance on the subsequent exercise. This was

also to ensure that participant's predictions were based upon the RT conditions that they had previously experienced. All exercises were supervised by one of the investigators who observed the participants whilst they performed the exercise without verbal encouragement so as to ensure consistency across participants. The investigator counted repetitions in their head and then noted these without the participant's knowledge. Prior to beginning each exercise participants were asked to consider the current load they were training with and to provide a prediction of the number of repetitions they could complete before reaching MF. Participants were informed that this was defined as the number of repetitions performed with the current load whilst continuing to the point where, despite the greatest effort and attempting to do so, they could not complete the current repetition (i.e., what repetition number they thought they would reach MF on). Participants were also asked to report their current training goals.

### Statistical analysis

Agreement between predicted and actual repetitions to MF was examined using standard error of measurement (SEM) and 95% confidence intervals (CI) in order to provide an absolute indication of the agreement between the variables. This was performed for each exercise. Calculations were performed using Microsoft Office Excel 2013 (Microsoft Corporation, Redmond, WA, USA) and spreadsheets for analysis of validity by *Hopkins (2015)* were used. Actual repetitions were considered the 'criterion' and predicted repetitions were considered the 'practical'.

## RESULTS

Descriptive statistics suggested that on average participants underpredicted the number of repetitions they could perform to MF (Table 1). For the combined sample SEMs (95%CIs) were 2.91 (2.61–3.30) for the chest press, 2.64 (2.36–2.99) for the elbow flexion, 3.38 (3.02–3.83) for the leg press, 2.95 (2.64–3.35) for the pulldown, 2.71 (2.43–3.08) for the seated row, and 3.36 (3.00–3.80) for the sit-up. SEMs and 95%CIs are reported in Table 2 for each exercise and group. There was a tendency towards improved accuracy in predicting actual repetitions to MF with greater experience across most exercises evidenced by reduced SEMs and narrower ranges between upper and lower 95%CIs. The training goal of muscular hypertrophy was reported with the highest frequency in the combined participant sample and all groups. Table 3 shows the training goals for each group by frequency.

## DISCUSSION

The current study examined the ability of participants to predict the number of repetitions they could perform to MF, with a given load, across a number of exercises and range of levels of experience. It was anticipated that participants would not be perfectly accurate in predicting actual repetitions to MF, in spite of performing exercises and using loads with which they were familiar. It was also hypothesised that there would be increased accuracy with greater RT experience. Descriptive data suggested participants on average

**Table 1  Descriptive data (mean ± SD) for each exercise and group.**

| | Combined | Orientation | Beginner | Experienced | Advanced | Expert |
|---|---|---|---|---|---|---|
| **Chest press** | | | | | | |
| Predicted | 12.38 ± 3.45 | 15.40 ± 2.77 | 14.86 ± 3.00 | 14.00 ± 3.46 | 11.14 ± 2.55 | 10.48 ± 2.86 |
| Actual | 14.21 ± 5.52 | 20.47 ± 6.36 | 18.76 ± 6.78 | 15.86 ± 4.16 | 12.17 ± 3.39 | 10.93 ± 3.17 |
| **Elbow flexion** | | | | | | |
| Predicted | 11.53 ± 2.99 | 15.40 ± 2.77 | 13.38 ± 2.85 | 12.67 ± 3.07 | 10.60 ± 2.06 | 9.60 ± 1.58 |
| Actual | 12.48 ± 4.32 | 18.20 ± 4.95 | 16.57 ± 4.65 | 13.14 ± 2.67 | 10.74 ± 2.36 | 9.81 ± 2.11 |
| **Leg press** | | | | | | |
| Predicted | 12.50 ± 3.41 | 15.40 ± 2.77 | 14.95 ± 3.14 | 13.67 ± 3.65 | 11.45 ± 2.62 | 10.71 ± 2.79 |
| Actual | 16.40 ± 6.29 | 23.87 ± 7.04 | 22.19 ± 5.75 | 17.62 ± 4.50 | 14.52 ± 4.13 | 12.10 ± 3.79 |
| **Pulldown** | | | | | | |
| Predicted | 12.48 ± 3.47 | 15.40 ± 2.777 | 15.27 ± 3.13 | 13.29 ± 3.36 | 11.55 ± 2.93 | 10.57 ± 2.77 |
| Actual | 13.81 ± 5.19 | 20.27 ± 6.92 | 17.86 ± 5.76 | 14.57 ± 4.04 | 11.95 ± 2.81 | 10.95 ± 2.83 |
| **Seated row** | | | | | | |
| Predicted | 12.38 ± 3.45 | 15.40 ± 2.77 | 14.86 ± 3.00 | 14.00 ± 3.46 | 11.14 ± 2.55 | 10.48 ± 2.86 |
| Actual | 14.09 ± 5.29 | 19.67 ± 5.38 | 18.33 ± 5.80 | 15.81 ± 4.24 | 12.26 ± 3.60 | 10.93 ± 3.48 |
| **Sit-up** | | | | | | |
| Predicted | 14.62 ± 3.05 | 15.93 ± 2.40 | 16.43 ± 2.62 | 15.71 ± 2.99 | 13.88 ± 2.92 | 13.43 ± 3.16 |
| Actual | 16.99 ± 3.65 | 18.73 ± 5.19 | 17.71 ± 3.99 | 17.76 ± 2.02 | 17.31 ± 3.23 | 15.29 ± 3.39 |

**Table 2  SEMs and 95%CIs for each exercise and group.**

| | Orientation | Beginner | Experienced | Advanced | Expert |
|---|---|---|---|---|---|
| Chest press | 4.33 (3.14–6.97) | 4.00 (3.04–5.84) | 2.58 (1.96–3.76) | 1.91 (1.57–2.45) | 1.57 (1.29–2.00) |
| Elbow flexion | 4.30 (3.12–6.93) | 6.39 (4.86–9.33) | 1.51 (1.15–2.20) | 1.95 (1.60–2.50) | 1.27 (1.04–1.62) |
| Leg press | 4.98 (3.61–8.03) | 3.37 (2.56–4.92) | 3.07 (2.34–4.49) | 2.49 (2.05–3.19) | 1.73 (1.42–2.21) |
| Pulldown | 4.15 (3.01–6.68) | 3.89 (2.96–5.68) | 2.49 (1.89–3.64) | 1.83 (1.50–2.34) | 1.35 (1.11–1.73) |
| Seated row | 4.51 (3.27–7.27) | 3.50 (2.66–5.11) | 2.28 (1.73–3.33) | 2.06 (1.69–2.63) | 1.71 (1.40–2.19) |
| Sit-up | 5.13 (3.72–8.26) | 4.08 (3.10–5.96) | 2.06 (1.57–3.01) | 2.87 (2.36–3.68) | 2.29 (1.88–2.94) |

underpredicted the number of repetitions they could perform to MF, though the average difference was reduced with greater experience. The SEMs indicated that participants indeed were not perfectly accurate at predicting repetitions to MF with SEMs for the combined sample ranging from 2.64 to 3.38 repetitions. In contrast with the descriptive data, SEMs suggested this was the case even for groups with greater experience, although there did still appear to be an improvement in accuracy with greater experience across most exercises. Considering the predominant training goal reported by the participants in this study (muscular hypertrophy), a less than perfectly accurate ability to predict repetitions to MF may have implications for achieving this goal.

Training to MF involves giving a maximal effort and is also anecdotally associated with higher discomfort. The less than perfectly accurate predictive ability reported herein may be a result of participants anchoring their prediction based upon discomfort. As we have recently noted in several papers (*Steele, 2014*; *Steele et al., 2017b*; *Steele et al., 2017a*), and

**Table 3  Participants training goals by frequency.**

| | Combined (n = 141) | Orientation (n = 15) | Beginner (n = 21) | Experienced (n = 21) | Advanced (n = 42) | Expert (n = 42) |
|---|---|---|---|---|---|---|
| Fitness | 6 (4.3%) | 1 (6.6%) | 2 (9.5%) | 2 (9.5%) | 0 | 1 (2.4%) |
| Maintenance | 3 (2.1%) | 0 | 1 (4.8%) | 0 | 1 (2.4%) | 1 (2.4%) |
| Muscular definition | 8 (5.7%) | 0 | 1 (4.8%) | 1 (4.8%) | 2 (4.8%) | 3 (7.1%) |
| Muscular hypertrophy | 107 (75.9%) | 10 (66.6%) | 14 (66.6%) | 14 (66.6%) | 36 (85.7%) | 33 (78.6%) |
| Strength | 4 (2.8%) | 0 | 1 (4.8%) | 0 | 1 (2.4%) | 2 (4.8%) |
| Weight loss | 13 (9.2%) | 4 (26.6%) | 2 (9.5%) | 4 (19.0%) | 1 (2.4%) | 2 (4.8%) |

as have others, differentiation between perceptions of effort and discomfort are important (*Abbiss et al., 2015*; *Marcora, 2009*; *Smirnaul, 2012*) particularly within RT (*Steele, 2014*; *Steele et al., 2017b*). In studies using traditional rating of perceived exertion scales higher ratings are given, despite conditions being controlled by supposedly training to MF, with lower loads for lower body exercise (*Shimano et al., 2006*), as set volume increases (*Silva et al., 2014*), with increased volume-load (*Pritchett et al., 2009*), and with increased work rate (*Hiscock et al., 2016*; *Hiscock, Dawson & Peeling, 2015*) supporting that participants may have expressed their feelings of increasing discomfort (*Steele, 2014*; *Steele et al., 2017b*; *Steele et al., 2017a*).

In some studies there have been attempts to differentiate between *effort* and *discomfort* during RT. Though participants appear able to report different values for each, there is a similar pattern for both responses. *Hollander et al. (2003)* and *Hollander et al. (2008)* found that, though effort is typically reported as being higher than discomfort (the authors used the term pain) under a range of RT conditions (different loads and muscle actions), both respond in a similar pattern. Such a relationship may be inherent; however, perception of effort is independent from afferent feedback mechanisms (*Marcora, 2009*). This would seem to disagree with observations of higher perceived efforts under conditions known anecdotally to induce higher feelings of discomfort (e.g., fatiguing low load lower body exercise). It is possible that participants were either consciously or unconsciously anchoring their effort and discomfort responses upon one another. When instructed to differentiate the two, participants are able to do so during RT (*Steele et al., 2017b*; *Fisher, Ironside & Steele, 2016*; *Fisher, Farrow & Steele, 2017*). But there appears to be a tendency to anchor one upon the other without such instruction.

Anchoring of perception of effort upon discomfort thus may have implications for whether a person is truly training to, or close to enough to, MF. Another point to consider is that participants in this study likely based their prediction upon prior experience of training whilst unsupervised as most persons train in this manner. Thus, the not perfectly accurate predictive ability of participants in this study might reflect that under unsupervised conditions participants are not reaching MF during training despite thinking that they may be, possibly due to the discomfort associated with such training. As such, persons training alone may find difficulty in training to MF unless highly self-motivated. Numerous studies report that strength and body composition changes are poorer when participants train unsupervised versus training under supervision (*Coutts, Murphy & Dascombe, 2004*;

*Gentil & Bottaro, 2010*; *Mazzetti et al., 2000*). When participants self-select RT load they often choose to train with lower loads than those recommended (*Elsangedy et al., 2013*; *Glass & Stanton, 2004*) and, considering the typical ranges of repetitions performed to MF by trainees at these loads (*Shimano et al., 2006*), are likely not training anywhere close to MF. Indeed, the RPE reported when participant's self-select load and repetition range, in addition to trainer observation, support this (*Glass & Stanton, 2004*). Instead, under supervision participants are more likely to train with heavier loads but also to report higher RPE (*Ratamess et al., 2008*). In fact it has been suggested that the poorer adaptations as a result of unsupervised training may be due to participants not training with sufficient proximity to MF and thus with lower effort (*Gentil & Bottaro, 2010*).

Evidently there may be implications for whether a person is able to achieve their training goals if they are unable to accurately perceive whether they are training to true MF or not. However, as noted there is disagreement within the literature as to whether performing RT to MF is indeed desirable and further that the consideration of MF in a dichotomous fashion (i.e., people training 'to MF' or 'not to MF') renders difficulty in understanding the nature of sub-maximal efforts during RT (*Steele et al., 2017a*). As a result, RIR scales have been developed to be used in controlling sub maximal effort in RT as an improvement upon the typical %1RM and traditional RPE based approaches (*Hackett et al., 2012*; *Hackett et al., 2016*; *Zourdos et al., 2016*). The results reported here are in agreement with other research (*Hackett et al., 2012*; *Hackett et al., 2016*) that participants are likely not perfectly accurate at predicting the number of repetitions they can perform to MF and thus suggest there may be reason to question the value of RIR scales. The use of RIR scales assumes a trainee is able to accurately predict the number of repetitions they could perform to MF. However, if a trainee is not perfectly accurate at making such a prediction then it is likely that they will be systematically training with a lower than desired effort level which may impact upon their adaptation to RT.

For untrained persons this may not be of considerable practical concern. In this case, even when using a lower than intended effort during RT on an individual set, cumulative fatigue can be induced by increased volume resulting in an increased effort, and thus closer proximity to MF, in later sets (*Fisher, Blossom & Steele, 2016*; *Giessing et al., 2016b*).

However, experience may a play role in a trainee's ability to predict proximity to MF and indeed the results reported here support this notion. There was a relationship between the level of experience of participants and the SEMs and width of 95%CIs found, with the most experienced group underpredicting by ∼1–2 repetitions compared with the least experienced underpredicting by ∼4–5 repetitions. *Hackett et al. (2012)* found experienced trainees ($8 \pm 3$ years RT experience) were initially not perfectly accurate at predicting repetitions to MF using the RIR scale over the first 1–2 sets (mean difference ranging 0.8–1.9 repetitions). However, on average accuracy improved in later sets. This suggests that, similar to our findings, even experienced trainees are still not perfectly accurate at predicting repetitions to MF, yet acute practice/experience appears to improve predictive ability. Further supporting the effect of experience, *Zourdos et al. (2016)* found that their RIR scale reflected more experienced lifters giving a more accurate estimation of their effort, particularly when using heavier loads, based upon average repetition velocities.

Thus the novice trainees in their study likely overestimated their effort and therefore were likely underpredicting how many repetitions away from MF they were.

Though increased experience would appear to increase predictive ability experienced trainees still under estimate by ∼1–2 repetitions. Thus using sub-maximal effort based RT prescriptions based upon RIR scales will result in most training at a lower than intended effort. For trained persons this may have a bigger impact upon adaptations. When attempting to stop a set of repetitions at a set end-point corresponding to a 'self determined repetition maximum' (where the participants were instructed to stop at the point they predicted MF on the next repetition) strength and hypertrophic outcomes may be sub-optimal (*Giessing et al., 2016a*). Thus, the use of 'intuitive' approaches that involve a person's ability to accurately predict the number of repetitions they could perform to MF may be questionable as an approach to prescribing and controlling effort in RT. However, the use of self determined repetition maximum based training compared with training to MF in experienced participants has only been examined with use of single set approaches (*Giessing et al., 2016b*). As accuracy of predictive ability improves with multiple sets of an exercise (*Hackett et al., 2012*) then RIR scales may have more utility in multiple set RT programs. Indeed, similarly to in untrained populations (*Fisher, Blossom & Steele, 2016*; *Giessing et al., 2016b*), even if training with a systematically lower than intended effort, the use of multiple sets, and thus the accumulation of fatigue, combined with improved predictive ability, makes it likely that in later sets trainees would be closer to achieving desired intensities of effort. As such, though even 'expert' participants in this study underpredicted by ∼1–2 repetitions, it is likely that this represents an acceptable degree of error if RIR scale based approaches to training are being utilised in multiple set routines. However, it seems as though even this degree of error has implications when using single set routines and as such predictive ability appears unacceptable for this approach.

It is worth considering the strengths and limitations of the present study. Firstly the present study was able to recruit a large sample size sufficient for examining validity/agreement between different measures (*Hopkins, 2000*). Due to this we were also able to sub group into a range of different experience levels. However, in order to achieve this large sample, participants were recruited from a private facility and testing conducted at this facility. This meant participants performed the testing using their current training equipment and load and, based upon the average repetitions, relative loads typically increased with experience. As such, the effects of experience level on the SEMs reported may be confounded as a result of differing ability to predict repetitions to MF when training using heavier or lighter loads. Greater predictive ability may therefore occur with heavier loads (*Zourdos et al., 2016*; *Helms et al., 2017*). This may be reflective of the conflation between effort and discomfort described above as greater perceived discomfort occurs with lower load RT (*Fisher, Ironside & Steele, 2016*; *Fisher, Farrow & Steele, 2017*). Future research should examine the impact that manipulation of other RT variables such as load, and its interaction with perceived discomfort, has upon ability to accurately predict repetitions to MF. A final limitation could be that we asked participants to predict the number of repetitions they could perform to MF prior to the execution of the exercise. Prior studies have asked participants during the execution of the set (*Hackett et al., 2012*;

*Hackett et al., 2016*) and thus participants may be able to make better predictions during the gestalt experience of actually performing the exercise. However, it should be noted that in studies where participants have attempted to stop one repetition prior to MF sub optimal adaptations have still been reported (*Giessing et al., 2016a*; *Giessing et al., 2016b*). Also, we did not control when the participant's penultimate training session prior to the testing sessions were, nor did we control and match other factors such as time of day, diet, sleep, etc. As such, there is still scope for further work to identify what factors may positively or negatively impact upon a person's predictive ability in performing repetitions to MF.

## CONCLUSION

Management of effort within RT by manipulation of whether a trainee reaches MF or not is a common approach by trainees and practitioners. Effort, and thus proximity to MF, may have implications for the optimisation of adaptations, in particular hypertrophy which for most commercial gym attendees is the most common training goal. Recently, RIR scales have been promoted as a means of controlling this and represent an improvement on %1RM and traditional RPE scales. However, they assume the trainee can make accurate predictions regarding their ability to perform repetitions to MF. The findings of the present study reveal that ability to predict repetitions to MF is not perfectly accurate amongst most trainees. However, there may be some increase in predictive ability with greater RT experience.

These results have implications regarding training adaptations from RT as most persons train unsupervised and thus are likely not training to actual MF in their current training programs. Further, for those not employing MF in their training but instead using sub-maximal efforts based upon proximity to MF, it is likely that they are systematically training with a lower than intended effort. These results suggest that RIR scales should be used with caution in most trainees. It appears that experience may improve a trainee's ability to predict repetitions to MF and therefore RIR scales may be more appropriate for experienced trainees. Lastly, these results apply to single set applications of RT. Prior research suggests with multiple sets predictive ability increases. As such, RIR scales may have the greatest utility in experienced trainees using multiple set RT programs.

### Funding
The authors received no funding for this work.

### Competing Interests
The authors declare there are no competing interests.

### Author Contributions
- James Steele analyzed the data, wrote the paper, prepared figures and/or tables, reviewed drafts of the paper.

- Andreas Endres and Jürgen Giessing conceived and designed the experiments, performed the experiments, reviewed drafts of the paper.
- James Fisher and Paulo Gentil analyzed the data, reviewed drafts of the paper.

## Human Ethics

The following information was supplied relating to ethical approvals (i.e., approving body and any reference numbers):

Due to the nature of the study conducted the University of Koblenz-Landau Institutional Review Board does not require formal submission of ethics as no blood or tissue samples are being taken.

## Data Availability

The raw data has been provided as Data S1.

## Supplemental Information

Supplemental information for this article can be found online at http://dx.doi.org/10.7717/peerj.4105#supplemental-information.

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
