# Peer review of "Ability to predict repetitions to momentary failure is not perfectly accurate, though improves with resistance training experience"

_PeerJ, doi:10.7717/peerj.4105_

## Round 0.1 · original submission · Major Revisions

The three reviewers all find some merit with your study and written manuscript, but also highlight some major concerns they have. In particular, reviewer one questions some of your calculations; while reviewers two and three both suggest you have perhaps interpreted the data in an overly negative fashion. I therefore suggest you look carefully at the comments of the three reviewers when preparing your resubmission.

Reviewer 1 ·

Basic reporting

Please see below

Experimental design

Please see below

Validity of the findings

I believe the statistics need to be re-run.

Additional comments

The authors have investigated an interesting idea, however, I believe the authors may need to re-calculate all of their values. For example, it’s hard to believe that a mean difference of 4 repetitions corresponds to a standard error of 1. I downloaded the spreadsheet and used your data and I believe that you put the “predicted” in the criterion instead of putting the “actual reps” as the criterion. Please see below. Also, you may consider implementing the minimal difference (SD of difference score x 1.96) in addition to what you have. I believe that these changes will likely change some of the interpretation of the manuscript.

Just a couple of examples,
Beginner Chest Press
You have 1.94 (1.48, 2.84)
I believe it should be 4.0 (3.04, 5.84)

Leg Press Expert
You have 1.27 (1.05, 1.63)
I believe it should be 1.73 (1.42, 2.21)

I would also delete the figure, it adds nothing to the manuscript.

A few other things to consider:
Introduction: Resistance training is associated, at best, with a wide range of health benefits…this should be made quite clear. The Dankel study does not actually offer much support for “resistance training” as it found the strongest associations with the outcome of strength as opposed to the behavior. I would tone this down some

Was this IRB approved? This information should be put into the manuscript.

Reviewer 2 ·

Basic reporting

Quite a negative tone concerning the accuracy and use of repetitions in reserve (RIR) during resistance training. Therefore, the writing style does not seem to present a balanced argument of RIR for resistance training.

Experimental design

The investigation performed is not adequate for the aim of the study. The investigators did not have participants estimate repetitions in reserve (RIR) during the actual exercise used to assess accuracy in RIR.Therefore the results presented are highly questionable.

Validity of the findings

There are a lot of unknowns that affect the results reported in this study, but the major one relates to the flawed methodological design.

Additional comments

Overall this paper is written in quite a negative tone concerning the accuracy of using repetitions in reserve (RIR) for monitoring of resistance training effort. An error of approximately 1 repetition is excellent for training purposes and RIR is best method that can be used to monitor resistance training effort. For example, RPE during resistance training is poorly correlated with RIR (soon to be published paper) and the information derived from RIR is impossible to fully decipher. So RIR is the obvious best method to provide coaches and trainers with added information, but RIR should not be used to prescribe resistance training. The RIR is most useful for assessing progression (adaptations) and fatigue (predictor of overtraining and maladaptation).

The sample size used for this study is great but there is a major methodological flaw. The accuracy of the RIR can only be assessed with a trainer reporting their rating during the exercise in which they will then performed repetitions to momentary failure. This is because you need the trainer to be able to utilise the many physiological and psychological cues to make their estimation. It makes no sense to have trainers make an estimation based on their previous training experience. The number of repetitions performed to failure with a specific relative load is likely to change on a day to day basis. Factors such as sleep, diet, previous training, time of day, other stressors, motivation etc. will impact upon the repeatability of performing repetitions to failure at a set load. The participants in your study may not have been motivated by the assessor but they were under research conditions and probably confounded your results. Also, it is no surprise that you found more experienced trainers would be more accurate with RIR. I might even hypothesise that this was due to the experience RT greater training frequency. Did you assess when your participants performed their last training sessions prior to your study? Potentially this could have impacted some of the results.

Specific concerns:
Line 65: ‘applications of RT to maximise these outcomes …..’ does not make sense? The ‘application’ refers to putting RT into practice. I think you mean the RT prescription?
Line 106-108: Trained participants are good at predicting repetitions to momentary failure. What criteria are you using to make your statement? You could not be possibly thinking that trainers can develop an ability to predict RIR with 100% accuracy. It will always be dependent on the loads used and proximity to momentary failure.
Line 194: Never end sentence with a prepositional primer (also line 316).
Lines 232-233: Anchoring of perception…… interesting because the study participants were making estimates based on previous exercise experience. Seems inappropriate to have this discussion

·

Basic reporting

No comment, all areas of basic reporting meet standards.

Experimental design

No comment, all areas of experimental design meet standards.

Validity of the findings

Data is robust and statistically sound and controlled. Interpretation is the only area where revision is recommended; see general comments.

Additional comments

First I wish to thank the authors for their contributions to this field of study. This is an important body of evidence to inform this area of research, the writing is clear, the findings are well presented, and the methods are sound. Additionally, the authors are correct that the work by Hackett and colleagues is influenced by the fact that multiple sets to failure were performed in both their 2012 and 2016 studies. Thus, as correctly pointed out by the authors, previous sets would help to improve accuracy of subsequent sets, thus potentially inflating the statistical representations of accuracy in both studies. Thus, the approach of estimating how many repetitions the lifter believes they can accomplish before ever beginning a set, and then comparing that estimation to the actual performance is novel.

That said, there are some revisions needed to ensure an accurate comparison of this data set is made to that of the existing literature.

In the title, and lines 41 (abstract), 107, 197, 202, 235, 261, 264, 276, 278, and the 318, the use of the word “poor” should be amended to a less subjective term. I would advise “not perfectly accurate” as this isn’t a debatable description, given the subjects in this study were unable to predict with complete accuracy how many repetitions they could perform to failure with a given load. The reason I believe this is important is the authors cite Hackett 2012 and 2016, and refer to the accuracy as “poor”, while Hackett and colleagues themselves in 2012 referred to the same accuracy as “high” and then as “quite good” in 2016 in the discussion sections of both papers. Specifically in their 2012 article, they state the following “Despite the differences between the estimated- and actual- repetitions-to-failure for the earlier sets of exercise, the difference was approximately only one repetition, indicating that participants only slightly underestimated this.” The use of “only slightly underestimated” contrasts the authors’ use of the word “poor” to highlight my point.

Both Hackett and colleagues and the present authors use a subjective description for the objective distance to failure achieved by their subjects. Neither can be said to be “correct” as it is a subjective determination, however, the authors can use less subjective terminology. Thus, please change all instances of “poor” that I highlighted above to “not perfectly accurate”, or similar, or when in doubt, simply state the actual distance from failure quantitatively as the number of repetitions.

The limitations the authors refer to when citing Hackett 2016 in lines 111-116 need to be revised as they contradict the methods of Hackett 2016. As it is stated in the methods of Hackett 2016 “Subjects aimed to perform 5 sets per exercise, but if concentric failure occurred prior to reaching 10 repetitions for a set, the testing for the exercise ceased.“ Additionally, in the beginning of the results section of Hackett 2012 it is stated “For both the bench press and squat, the estimated- and actual-repetitions-to-failure for set 5 was 0 for all participants. Therefore, data from set 5 were excluded from the ANOVA.” Based on these two quotes, the authors should revise the stated limitations of the data by Hackett and colleagues as it doesn’t appear that sets were continued if failure was reached at or before repetition 10, and if failure was reached at repetition 10, the estimation of 0 repetitions remaining was not included in the analysis.

Finally, I think the authors do a fantastic job pointing out the issue of an a-priori assumption of accuracy and of the short comings of RIR as a resistance training prescriptive tool. However, on balance, I think the authors should also point out some of the benefits of RIR based load prescription. One, that the authors already point out, is that it is perhaps more appropriate for strength athletes and bodybuilders vs untrained individuals. Indeed, the Hackett 2012 paper showing ~1 repetition away from accuracy was on bodybuilders, and almost the entirety of work by Zourdos and Helms is on competitive powerlifters. I think that should be specifically pointed as both a limitation for general population users, and potentially a benefit for strength and physique athletes who are much less likely to self-select loads that will not be challenging or lead to adaptation.

Also on balance, please point out what the RIR approach improves on. As the authors pointed out in lines 210-213, studies using traditional Borg RPE sometimes report RPE values in the 6-9 out of 10 range (depending on the study) even when training to muscular failure. Thus, as was shown in Hackett 2012, RIR is likely an improvement Borg RPE for resistance training. Additionally, the authors also point out in lines 87-88, there is large variation in the number of repetitions that can be performed at the same relative load (% 1RM) between individuals. In fact, these differences can be quite large; in endurance athletes compared to strength athletes, on average, a difference of 22 reps at 70%, 8 reps at 80% and ~4 reps at 90% of 1RM https://www.ncbi.nlm.nih.gov/pmc/articles/PMC4042664/. Thus, it would seem that even with the inaccuracy found in the present study, RIR is likely to provide a load closer to the intended difficulty than percentage 1RM based load prescription as well.

To conclude, excellent work, this data is an important contribution collected in a novel manner, and I would recommend the following amendments in order of importance as:

1. Changing the subjective wording of poor to “not perfectly accurate” or similar, or just stating distance from failure when referring to the present and previous studies (by Hackett)
2. Amending the limitations stated about the Hackett paper.
3. Including the fact that despite its limitations, RIR is likely a better alternative than %1RM or Borg RPE for load prescription in resistance training.
4. Adding additional detail to the greater utility for trained individuals, specifically strength and physique athletes as the subjects in papers by Helms and Zourdos and Hackett 2012, were powerlifters and bodybuilders respectively.

---

## Round 0.2 · Minor Revisions

The authors are to be congratulated for making virtually all of the requested changes to the manuscript. There are just a few very minor amendments requested before this paper can be accepted for publication.

Reviewer 1 ·

Basic reporting

The authors have substantially improved the manuscript…I have only a few more points
1) you use a comma instead of a “.” In the Table for chest press…please be consistent

2) I believe some of the references you deleted in text still show up in your reference list

3) you say repeatedly that people are “not perfectly accurate”…but you never really mention what would be acceptable? What would be considered good..1 rep? It seems unlikely that you’d ever predict it perfectly?

Experimental design

See above

Validity of the findings

see above

Additional comments

see above

Reviewer 2 ·

Basic reporting

No comment

Experimental design

No comment

Validity of the findings

No comment

·

Basic reporting

no comment

Experimental design

no comment

Validity of the findings

no comment

Additional comments

I wish to thank the authors for considering all of my proposed revisions. I have no further suggested revisions, and I humbly suggest the paper is now improved. Well done!

---

## Round 0.3 · accepted · Accept

You are all to be congratulated for the dedication and thoroughness you have displayed in attending to the constructive criticisms of the three reviewers. On this basis, I am happy to confirm that your manuscript has been accepted for publication in PeerJ.

Reviewer 1 ·

Basic reporting

The authors have adequately addressed my concerns.

Experimental design

The authors have adequately addressed my concerns.

Validity of the findings

The authors have adequately addressed my concerns.

Additional comments

The authors have adequately addressed my concerns.